# Adoption of AI writing tools among academic researchers: A Theory of Reasoned Action approach

**Mohammed A. Al-Bukhrani**[1]*, **Yasser Mohammed Hamid Alrefaee**[2], **Mohammed Tawfik**[3]

**1** Department of Accounting, Faculty of Administrative Sciences, Albaydha University, Albaydha, Yemen, **2** English Department, Faculty of Education, Albaydha University, Albaydha, Yemen, **3** Faculty of Computer and Information Technology, Sana'a University, Sana'a, Yemen

* mabuazzam2016@gmail.com

**Data Availability Statement:** All relevant data are within the manuscript and its Supporting information files.

**Funding:** The author(s) received no specific funding for this work.

## Abstract

This research explores the determinants affecting academic researchers' acceptance of AI writing tools using the Theory of Reasoned Action (TRA). The impact of attitudes, subjective norms, and perceived barriers on researchers' intentions to adopt these technologies is examined through a cross-sectional survey of 150 researchers. Structural Equation Modeling (SEM) is employed to evaluate the measurement and structural models. Findings confirm the positive influence of favorable attitudes and subjective norms on intentions to use AI writing tools. Interestingly, perceived barriers did not significantly impact attitudes or intentions, suggesting that in the academic context, potential benefits may outweigh perceived obstacles to AI writing tool adoption. Contrarily, perceived barriers do not significantly affect attitudes and intentions directly. The TRA model demonstrates considerable explanatory and predictive capabilities, indicating its effectiveness in understanding AI writing tool adoption among researchers. The study's diverse sample across various disciplines and career stages provides insights that may be generalizable to similar academic contexts, though further research with larger samples is needed to confirm broader applicability. Results offer practical guidance for tool developers, academic institutions, and publishers aiming to foster responsible and efficient AI writing tool use in academia. Findings suggest strategies such as demonstrating clear productivity gains, establishing AI Writing Tool programs, and developing comprehensive training initiatives could promote responsible adoption. Strategies focusing on cultivating positive attitudes, leveraging social influence, and addressing perceived barriers could be particularly effective in promoting adoption. This pioneering study investigates researchers' acceptance of AI writing tools using a technology acceptance model, contributing to the understanding of technology adoption in professional contexts and highlighting the importance of field-specific factors in examining adoption intentions and behaviors.

**Competing interests:** The authors have declared that no competing interests exist.

## Introduction

Artificial intelligence (AI) tools for generating and enhancing text have advanced rapidly in recent years. Large language models such as GPT-3 and specialized writing assistants now support various professional workflows [1–4]. In academic research, AI tools from Anthropic, startups, and major publishers assist with tasks such as summarizing papers, providing writing suggestions, catching errors, answering questions, citing sources, and augmenting peer review [4].

The emergence of these AI writing tools has sparked significant interest and debate within academia. While comprehensive statistics on adoption rates are still emerging, recent studies suggest a growing interest in and use of these tools among researchers. For instance, a survey conducted shortly after the release of ChatGPT found that 22% of researchers had already used it in their work, with many employing it for writing assistance [5]. This rapid uptake indicates the potential impact of AI writing tools on academic practices.

However, despite the hype around these tools, real-world adoption of AI writing assistants remains limited. Surveys indicate skepticism about their effectiveness and use cases [6]. This disparity between awareness and adoption suggests that various factors influence researchers' decisions to incorporate these tools into their workflows.

The potential benefits of AI writing tools for academia are substantial. They can enhance productivity by streamlining the writing process, improve language quality for non-native English speakers, and even stimulate new ideas through text generation [1]. However, their use also raises concerns about authorship, academic integrity, and the potential homogenization of scholarly writing [4].

The Unified Theory of Acceptance and Use of Technology (UTAUT) suggests adoption depends on perceived usefulness, ease of use, social norms, self-efficacy, and barriers [7]. While research has applied UTAUT to predict adoption of digital innovations, understanding of these drivers for AI writing tools in professional contexts is lacking.

While UTAUT has been widely applied in recent AI adoption studies [8–10], our choice of TRA offers unique contributions to this research. TRA's focus on attitudes and subjective norms aligns closely with the academic context, where peer influence and individual perceptions significantly impact technology adoption [11]. Unlike UTAUT's broader scope, TRA's parsimony allows for a more focused examination of core psychological factors influencing adoption intentions, which is particularly valuable in the early stages of AI writing tool integration in academia. Recent studies have demonstrated TRA's effectiveness in explaining AI technology adoption, either standalone [12] or in combination with other models [13]. By applying TRA to AI writing tool adoption in academia, we aim to provide complementary insights to UTAUT-based research, offering a nuanced understanding of the attitudinal and normative factors specific to this context.

The Theory of Reasoned Action TRA provides a relevant framework to investigate adoption perceptions. It posits that adoption intentions are shaped by attitudes towards innovations and subjective norms around usage [14]. These factors, along with external variables such as barriers, can predict technology use [15]. Since AI writing tools represent a shift in practices for knowledge professionals [16], TRA is suitable for examining the factors influencing their adoption.

Building on this theoretical foundation, our study applies the TRA framework to address the research gap in understanding AI writing tool adoption among academic researchers. By leveraging TRA's emphasis on attitudes and subjective norms, we aim to provide novel evidence on the key cognitive, social, and barrier-related factors shaping researchers' willingness to use AI for academic writing. This approach allows us to delve deeper into the psychological

and social dynamics specific to the academic context, offering insights that complement and extend existing research using broader models such as UTAUT. Specifically, we seek to answer the following research questions:

1. How do attitudes towards AI writing tools and perceived subjective norms influence researchers' intentions to adopt these technologies?

2. What role do perceived barriers play in shaping attitudes and adoption intentions towards AI writing tools?

3. To what extent does the TRA model explain and predict AI writing tool adoption among academic researchers?

An original questionnaire was deployed to over 150 researchers across disciplines, seniority levels, and AI familiarity levels (see S1 Appendix). The survey measured intentions to use AI writing tools, attitudes, perceived norms, and barriers. Structural Equation Modeling (SEM) was employed to evaluate the measurement and structural models, exploring the hypothesized TRA relationships. The results offer implications for developers, institutions, and publishers on evidence-based ways to support AI writing tool adoption. This includes addressing access, skills, ethics, and citation norms. By focusing on the academic community, the analysis provides tailored insights to guide the development of AI tools that enhance scholarly productivity and integrity.

This study makes three key contributions: It is the first known research investigating AI writing assistant adoption perceptions among academic researchers using a technology acceptance model; The findings validate TRA relationships in this context while highlighting unique barriers facing researchers; a Actionable recommendations are provided for institutions, publishers, and vendors to promote responsible and effective usage.

Further research can build on these adoption factors using larger samples, usage data, and workflow investigations. However, this work is an important first step in elucidating the key perceptions underlying researchers' willingness to use AI tools for generating and critiquing manuscript ideas. The rest of the paper reviews the relevant literature, details the methodology, presents the empirical results, and discusses the findings, implications, and future directions.

The remainder of this paper is structured as follows: First, we present a comprehensive review of the relevant literature, focusing on the Theory of Reasoned Action and its application to technology adoption in academic contexts. Next, we detail our research methodology, including the development of our survey instrument and our approach to data analysis using Structural Equation Modeling. We then present our empirical results, followed by a discussion of our findings and their implications for various stakeholders. Finally, we conclude by summarizing our key contributions, acknowledging limitations, and suggesting directions for future research in this rapidly evolving field.

## Literature review

### Emerging AI writing technologies

The emergence of AI writing technologies, such as OpenAI's ChatGPT, has sparked discussions on their potential impact on academia and scholarly publishing [17]. These tools are considered potential aids for improving the quality of scholarly content and streamlining the writing process [18]. AI writing tools have been proven to provide substantial advantages in educational environments [19], and educators are urged to utilize these tools to improve the calibration and productivity of academic work [20]. AI tools are anticipated to improve and simplify the scientific publishing process [21]. Nevertheless, there are apprehensions regarding

the possible ramifications of AI writing tools on the genuineness and reliability of scholarly endeavours [4]. It is important for the scientific community to actively participate in conversations regarding the possible outcomes of these technologies [22], and to prioritize the ethical utilization of these tools [23]. AI writing technologies are anticipated to have a growing impact on the future of scholarly communication and research processes as they continue to advance.

## Theory of Reasoned Action and related technology adoption models

The Theory of Reasoned Action (TRA), established by [24], is a regularly utilized conceptual framework for comprehending and forecasting human behavior. According to the TRA, an individual's purpose in engaging in a particular conduct is the primary factor that influences and predicts their actual behavior. Behavioral intentions are influenced by attitudes towards activity and subjective norms [25]. Attitudes pertain to an individual's satisfactory or unsatisfactory assessments of engaging in a certain conduct, whereas subjective norms represent the perceived societal influence on either participate or abstain from the behavior [26].

The TRA has been effectively utilized in several fields, such as the adoption of technology [27], health behaviors [28], and environmental conservation [29], demonstrating its robustness and versatility in predicting and explaining human behavior. [30] found that the decomposed TRA provided a fuller understanding of behavioral intention, while [31] further supported the TRA, highlighting the importance of trust in understanding Internet banking behavior. [32, 33] both found that TRA was a strong predictor of purchase intention and technological innovation adoption, respectively. [34] applied TRA to the acceptance of Green Information Technology, finding that external factors significantly influenced attitude.

Building on the foundation of the TRA, other models have been developed to elucidate and forecast user acceptance of novel technologies. The Technology Acceptance Model (TAM), introduced by, introduced by [27], is a modified version of the TRA designed exclusively for the information systems domain. TAM identifies two crucial aspects that affect individuals' intentions to adopt a technology: perceived usefulness and perceived ease of use.

Another significant model is the Unified Theory of Acceptance and Use of Technology (UTAUT), formulated by [35], which combines components from many models of technology adoption, such as the TRA and TAM, to offer a holistic framework for comprehending the acceptance of technology. The UTAUT framework highlights four fundamental factors that influence individuals' intents to adopt something: The factors that influence performance are performance expectancy, effort expectancy, social influence, and facilitating conditions.

These models have been widely applied in research on technology implementation, and their associated measures, such as the Technology Acceptance Scale [27] and the UTAUT questionnaire [35], have been validated across various contexts. [36] found that both TAM and UTAUT were effective in predicting smartphone usage intentions in Korea, while [37, 38] applied TAM and UTAUT respectively to the adoption of ICT in education and universities. [39, 40] both extended UTAUT, with the former focusing on Microsoft Project Management Software and the latter proposing the "Five Forces of Technology Adoption" framework. [41] combined TAM with the Theory of Technology Readiness to predict the acceptance of Online Attendance Systems in higher education.

Recent studies have demonstrated the continued relevance of these models in understanding AI adoption. For example [8], employed a fusion of UTAUT and the Expectation-Confirmation Model (ECM) to investigate the acceptance of AI Chatbots among graduate students in China. [9] applied UTAUT to explore AI adoption in healthcare, while [10] used it to analyze the intention to adopt AI in human resource recruitment.

However, some researchers have found value in using the TRA or combining it with other models for AI adoption studies. [13] simultaneously employed UTAUT and TRA to study the adoption of AI-enabled robo-advisors in fintech. Their results supported the importance of both UTAUT constructs and TRA-related factors in shaping user attitudes and adoption intentions. In the specific context of AI writing tools [12], successfully applied TRA to examine factors influencing users' intentions to switch to ChatGPT from traditional search engines. Lastly [42, 43], both extended TAM to include trust and consumer personality factors in the context of mobile marketing and smartphone usage.

Several studies have extensively examined the adoption of artificial intelligence (AI) tools in various contexts, with a particular emphasis on the Unified Theory of Acceptance and Use of UTAUT and TAM frameworks. [44, 45] employed the UTAUT framework to analyse the factors that influence the adoption of AI tools, including individual, technological, and environmental characteristics. [46] developed a measurement tool based on the TAM to assess the acceptance of AI-based assessments by learners. [47] compared the TAM, Theory of Planned Behaviour (TPB), UTAUT, and Value-Attitude-Model (VAM) to examine customer adoption of AI-based intelligent products. [45] further investigated the impact of AI aversion as a moderating factor, adding depth to our understanding of potential barriers to AI adoption. In the context of technology acceptance [33], emphasized the usefulness of the TRA. Specifically in the realm of AI-driven assessment [48], introduced theoretical frameworks based on the Technology adoption Model (TAM) to evaluate the level of adoption of AI-driven assessment among students and educators in eLearning, respectively.

Interestingly, in their comparison of various models [47], found that the Value-Attitude-Model (VAM) was most successful in explaining consumer acceptance of AI-based intelligent products. This finding highlights the importance of considering multiple theoretical perspectives when studying AI adoption. In the higher education context [49], UTAUT to examine AI adoption, further demonstrating the applicability of these models in academic settings.

Collectively, these studies suggest that the TRA, TAM, and UTAUT can all provide valuable insights into the acceptability of AI tools. The diversity of approaches and findings indicates the potential for further investigation in the field of research writing, particularly in understanding the unique factors influencing AI writing tool adoption in academia.

This study opts to employ the TRA as its primary theoretical framework, despite the widespread use of other technology acceptance models such as TAM and UTAUT in various situations. The TRA is particularly suitable for examining the adoption of AI writing tools among academic scholars for various reasons. It highlights the importance of attitudes and subjective norms in shaping behavioural intentions, as supported by [24]. This aligns with the current study's emphasis on researchers' perceptions and societal influence in the acceptance process. The TRA has demonstrated its robustness and flexibility in predicting and explaining human behavior in various fields, including technology adoption. It provides an efficient structure that allows for the integration of context-specific factors, such as perceived barriers, relevant to the unique challenges researchers may have while using AI writing tools. Recent successful applications by [13, 50] have proven TRA's effectiveness of TRA in explaining AI technology adoption, either standalone or in combination with other models.

Given these advantages and recent successful applications in AI contexts, the TRA provides a robust framework for examining the adoption of AI writing tools in academia. This study aims to provide a focused knowledge of the key elements that influence the acceptability of AI writing tools in the academic setting by examining the fundamental constructs of attitudes, subjective norms, and perceived hurdles within the TRA framework.

## Hypotheses development based on TRA

Drawing upon the TRA and its key constructs, we develop a set of hypotheses to investigate the factors influencing academic researchers' adoption of AI writing tools.

*H1*: *Attitudes towards AI writing tools positively influence researchers' intentions to use these tools.*

Researchers' positive evaluations of AI writing tools based on their perceived benefits and outcomes are expected to increase their intention to adopt these technologies [26, 27].

*H2*: *Perceived barriers negatively influence researchers' attitudes towards AI writing tools.*

Researchers' perceptions of potential barriers, such as technical difficulties, ethical concerns, or institutional restrictions, may negatively impact their attitudes towards AI writing tools [51, 52].

*H3*: *Perceived barriers negatively influence researchers' intention to use AI writing tools.*

In addition to influencing attitudes, perceived barriers may directly discourage researchers from intending to use AI writing tools [35].

*H4*: *Perceived subjective norms positively influence researchers' attitudes towards AI writing tools.*

The opinions and expectations of influential others in the academic community can shape researchers' evaluations and beliefs about AI writing tools [53].

*H5*: *Perceived subjective norms positively influence researchers' intentions to use AI writing tools.*

Social pressure and the perceived expectations of colleagues, supervisors, and the broader research community may directly influence researchers' intentions to adopt AI writing tools [24, 35]

*H6*: *Perceived subjective norms negatively influence perceived barriers to using AI writing tools.*

When influential others in the academic community express support for and encourage the use of AI writing tools, researchers may perceive fewer barriers to adoption [53]. By testing these hypotheses, this study aims to provide a comprehensive understanding of the factors influencing AI writing tool adoption among academic researchers, grounded in the TRA and related technology adoption models. The empirical findings will contribute to the advancement of technology adoption research in the context of AI-based tools and inform evidence-based strategies for their successful implementation in academic settings.

## Methodology

### Research design

This study employed a cross-sectional survey design with a quantitative approach to investigate the factors influencing academicians' adoption of AI writing tools.

### Sampling and data collection

The study focused on academic researchers from many disciplines and career stages as the target audience. The researchers used a purposive sampling strategy to pick participants who could offer valuable insights into the use of AI writing tools in academia [54]. Purposive sampling enables researchers to intentionally select participants based on particular features or experiences that are pertinent to the study aims [55].

Prior to data collection, this study received ethical approval from the Research Ethics Committee at Albaydha University. All participants provided informed electronic consent at the beginning of the online survey, which detailed the study's purpose and confirmed voluntary participation. Participants were informed that their data would be anonymized and used solely for research purposes. Those who did not consent could not proceed with the survey.

To ensure a comprehensive and representative sample, specific criteria were established for participant selection. These included academic roles ranging from doctoral candidates in their final year to full professors, representation from at least five major academic disciplines (STEM, social sciences, humanities, business/economics, and interdisciplinary programs), varying levels of familiarity with AI writing tools, balanced distribution across career stages, and institutional diversity encompassing different types of academic settings.

Following [56, 57], the sample size was determined using a priori power analysis Sample Size Calculator proposed by [58] for SEM Four laten variables measured by 23 observed variables (items) with an anticipated effect size of 0.3, a desired power level of 0.8, and a significance level of 0.05 [58, 59]. The analysis indicated a minimum sample size of 133 and a maximum of 166 participants to detect significant effects and ensure adequate model fit [60].

Participants were recruited through a combination of email invitations and WhatsApp [61]. Email invitations were the primary method, with addresses sourced from publicly available faculty and researcher directories of targeted institutions and academic networks such as ResearchGate and Academia.edu Additionally, WhatsApp messages were sent to relevant academic groups, and a snowball sampling technique was employed, asking initial participants to recommend colleagues who met the criteria, particularly those from underrepresented groups in academia. The invitation included a brief description of the study, the estimated time commitment, and a link to the online survey using google form. To encourage participation and reduce non-response bias, the invitation emphasized the importance of the research and the potential benefits for the academic community [62].

Data were collected using an online survey to facilitate wider reach and enable participants to complete the questionnaire at their convenience [63]. The survey was designed using a user-centric approach, incorporating clear instructions and a logical sequence of questions. Participants were provided with guarantees of secrecy and anonymity to encourage honest and direct feedback [64]. The data collection phase lasted for three weeks, with reminder emails sent to non-respondents to increase the response rate [65]. From 215 replies, 150 complete and appropriate responses were obtained, yielding a 70% response rate.

The final sample comprised researchers from diverse academic disciplines, career stages, and levels of familiarity with AI writing tools, ensuring a representative and heterogeneous study population. This diversity was reflected in the distribution of participants across the sampling criteria, with representation from various academic roles, disciplines, and career stages aligned with the predetermined quotas set in the sampling strategy.

## Survey instrument and measures

The questionnaire utilized in this investigation was constructed using the TRA framework [24] and modified from established scales employed in previous technology adoption studies. The survey measures various dimensions, including the intention to utilize AI writing tools, attitudes towards AI writing tools, perceived subjective norms, and perceived barriers. The process of adapting established scales to the context of AI writing tools in academia involved several key steps. First, each item from the original scales was evaluated for relevance to AI writing tools in academic settings. Next, relevant items were reworded to specifically mention AI writing tools while maintaining the core construct being measured. For example, a TAM

item "Using the system in my job would increase my productivity" was adapted to "Using AI writing tools would increase my productivity in academic writing." Additionally, new items were created to address unique aspects of AI writing tools in academia, such as ethical considerations and integration with existing academic workflows.

The metrics for assessing the inclination to utilize AI writing tools were modified from TAM developed by [27] and the Unified Theory of UTAUT defined by [35]. For instance, one item stated, "I intend to use an AI writing assistant the next time I work on a research paper, manuscript, or other academic document," with responses ranging from "Strongly Disagree" to "Strongly Agree." These studies have established the validity and reliability of the intention construct in predicting technology adoption behavior.

Attitudes towards AI writing tools were assessed using items adapted from [66], who investigated changes in beliefs and attitudes toward information technology usage. An example item asked respondents to rate whether "Using an AI writing assistant to help craft elements of my academic writing would be:" on a scale from "Very Bad Practice" to "Very Good Practice."

Perceived subjective norms were measured using items adapted from [35], who incorporated social influence as a key determinant of technology adoption in the UTAUT model. These items were rephrased to capture the influence of academic peers, researchers, and collaborators on the adoption of AI writing tools. For example, one item stated, "Many of my academic peers think using AI tools to enhance writing is a favorable practice."

Perceived barriers were assessed using items adapted from [67], who explored the factors influencing the adoption of AI technologies in education. These items were modified to address potential barriers specific to academic AI writing tools. These included items such as "I am concerned about the potential high costs of accessing quality AI writing tools and assistants."

In addition to the main constructs, the survey collected demographic information, such as career stage and academic discipline, to enable subgroup analysis and comparison. Familiarity with AI tools was measured using a single item adapted from [35], while prior usage and experience items were developed by the authors to capture specific aspects of AI writing tool adoption in the academic context.

The development of the survey instrument followed best practices in questionnaire design, such as using clear and concise language, providing unambiguous response options, and ensuring a logical flow of questions [65]. The adapted items were carefully reviewed and modified to maintain the integrity of the original constructs while ensuring relevance to the research context.

To establish face and content validity, the adapted questionnaire underwent a pre-test with a select group of eight researchers from diverse disciplines. Their feedback led to several important modifications. The term "AI writing tools" was more precisely defined at the beginning of the survey based on feedback about potential ambiguity. For the perceived barriers construct, the response options were changed from a general agreement scale to more specific descriptors (e.g., "Not at all a Barrier" to "Very Much a Barrier") based on pre-test suggestions. Two new items were added to the perceived barriers construct to address concerns raised about institutional policies and ethical implications that were not initially covered. Several items were rephrased to improve clarity and reduce potential misinterpretation, particularly in the perceived subjective norms section.

Several procedures were implemented beyond the use of reminder emails to ensure data quality and to handle incomplete responses. Key items were set as required to prevent accidental skipping.

The final questionnaire was tested with a select group of researchers and specialists that specialize in the areas of technology adoption and academic writing. Their feedback was incorporated to further refine the items and improve the overall clarity of the survey [68].

## Data analysis

Descriptive statistics, including frequencies, percentages, means, and standard deviations, were computed using SPSS (Version 28) to summarize the characteristics of the sample and important factors. The measurement model and structural model were evaluated using Smart-PLS (Version 3) to conduct SEM analysis [69]. The measuring model's reliability, convergent validity, and discriminant validity were evaluated using benchmarks such as Cronbach's alpha, composite reliability, AVE, and HTMT [70]. The structural model was evaluated for collinearity, path coefficients, explanatory power (R-squared), predictive power (Q-squared), and model fit using the standardized root mean square residual (SRMR) and bootstrap-based confidence intervals [71].

## Results

### Descriptive statistics

Table 1 presents an overview of the key sample characteristics, discipline representation, prior familiarity levels, and usage frequency for the 150 surveyed researchers. This enables the

**Table 1. Sample characteristics and AI writing tools usage.**

| Demographic | Category | Frequency | Percent |
|---|---|---|---|
| **Career Stage** | Grad Student/Post-doc/Junior Faculty | 120 | 80% |
| | Advanced Assistant/Associate Professor | 21 | 14% |
| | Senior Faculty/Professor | 9 | 6% |
| | Total | 150 | 100% |
| **Academic Discipline** | Physical Sciences (Physics, Chemistry, etc.) | 16 | 10.70% |
| | Social Sciences (Psychology, Sociology, etc.) | 6 | 4% |
| | Arts & Humanities | 46 | 30.70% |
| | Health Sciences | 8 | 5.30% |
| | Business/Management Fields | 42 | 28% |
| | Engineering/Computer Science | 32 | 21.30% |
| | Total | 150 | 100% |
| **Gender** | Male | 121 | 80.70% |
| | Female | 29 | 19.30% |
| | Total | 150 | 100% |
| **Familiarity with AI Tools** | Not at all familiar | 12 | 8% |
| | Slightly familiar | 42 | 28% |
| | Moderately familiar | 52 | 34.70% |
| | Very familiar | 36 | 24% |
| | Extremely familiar | 8 | 5.30% |
| | Total | 150 | 100% |
| **Prior Usage** | Never | 20 | 13.30% |
| | Once or Twice | 40 | 26.70% |
| | Monthly | 32 | 21.30% |
| | Weekly | 34 | 22.70% |
| | Daily | 24 | 16% |
| | Total | 150 | 100% |
| **Prior Experience** | Writing Suggestions | 73 | 48.70% |
| | Auto Grammar Checking | 83 | 55.30% |
| | Research Paper Editing | 42 | 28% |
| | Article Summarization | 38 | 25.30% |
| | ChatGPT | 90 | 60% |

assessment of exposure diversity and aids contextualization of AI writing tool adoption patterns discovered in subsequent analyses. The sample comprises graduate students, faculty ranging from junior to senior ranks and small number (6%) of emeritus professors, spanning varied career maturity. However, over 80% are in the early career stage, indicative of a left skew distribution towards less established researchers. In terms of disciplinary spread, representation hovers between 10–30% per category besides social sciences (4%). The plurality from arts, humanities (30.7%) and business/management (28%) fields aligns with likelihood of higher writing output needs. 25% have technical science and engineering backgrounds potentially influencing receptivity to AI solutions. Familiarity levels display polarization on one end 12 (8%) participants report zero prior awareness while on the other 8 (5%) claim extreme familiarity. Bulk range from slightly (42; 28%) to very (36; 24%) familiarity curves denoting uneven dispensation of preexisting tool awareness. Usage frequency adds further evidence of split adoption: 20 (13%) never users contrast with 24 (16%) incorporating assistance daily. Modal classes lie in intermittent (once/twice: 26.7%) or moderate weekly/monthly (21–22%) adoption. Finally, prior hands-on experience spanning basic to advanced aid types shows participation bifurcated between the no-use and direct-exposure segments.

Table 2 provides a more granular overview of variations in both direct hands-on experience and usage frequency with AI-enabled writing enhancements based on researchers' academic disciplines and career stages. This two-dimensional perspective enriches the understanding of adoption patterns beyond aggregate trends. In terms of prior first-hand experience, basic and widespread aid types display relatively higher exposure levels across categories—writing suggestions (33–60%), grammar checking (43–67%). However, scholars employing more advanced functionality like research paper editing (9–33%) or article summarization (0–29%) remain low implying narrow integration so far by pioneers. Significant ChatGPT awareness (47–74%) punctuates its explosive trial despite recency—aligned with consumer deployment

**Table 2. AI writing tools experience levels/prior usage of AI writing tools by academic discipline and career stage.**

| Demographic | Category | Prior Experience (*I have prior experience using the following AI writing tools or assistants?*) | | | | |
|---|---|---|---|---|---|---|
| | | Writing Suggestions | Grammar Checking | Research Editing | Article Summarization | ChatGPT |
| Academic Discipline | Physical Sciences | 43.80% | 43.80% | 18.80% | 12.50% | 56.30% |
| | Social Sciences | 33.30% | 50% | 16.70% | 0% | 50% |
| | Arts & Humanities | 45.70% | 58.70% | 26.10% | 19.60% | 65.20% |
| | Health Sciences | 37.50% | 50% | 25% | 0% | 50% |
| | Business/Management | 59.50% | 66.70% | 33.30% | 28.60% | 73.80% |
| | Engineering/Computer Science | 43.80% | 62.50% | 28.10% | 21.90% | 62.50% |
| Career Stage | Grad Student/ Post-doc/ Junior Faculty | 51.70% | 58.30% | 30.80% | 27.50% | 65% |
| | Advanced Assistant/ Associate Professor | 33.30% | 47.60% | 9.50% | 4.80% | 47.60% |
| | Senior Faculty/ Professor | 55.60% | 66.70% | 22.20% | 11.10% | 66.70% |
| **Prior Usage** (*The past year, how often have you used some form of AI technology to assist with your academic writing?*) | | | | | | |
| Academic Discipline | Physical Sciences | 13.3% Never <br> 16% Daily | | | | |
| | Social Sciences | 33.3% Never while 16.7% Daily | | | | |
| | Arts & Humanities | 13% Never while 13% Daily | | | | |
| | Health Sciences | 12.5% Never while 25% Daily | | | | |
| | Business/Management | 7.1% Never while 26.2% Daily | | | | |
| | Engineering/Computer Science | 21.9% Never while 12.5% Daily | | | | |
| Career Stage | Grad Student/Post-doc/Junior Faculty | 12.5% Never while 18.3% Daily | | | | |
| | Advanced Assistant/Associate Professor | 19% Never while 9.5% Daily | | | | |
| | Senior Faculty/Professor | 11.1% Never while 22.2% Daily | | | | |

models. By discipline, business, arts and engineering researchers demonstrate greater tool experience, unsurprising given extensive writing demands. But social science and health domains lag considerably (0–16% complex aid adoption), likely highlighting productivity necessity conditions. Across career trajectories, seniormost faculty use outpaces other tiers (23–67% relative to 10–58%) potentially tied to publishing pressures and technological acclimatization over time. Usage frequency narratives indicate peak polarization—13–21% never harness automation contrasting daily reliance for 12–26% across clusters. However, appetite persists, as illustrated by wider intermittent (monthly/weekly) users (22–26%). Distinct discipline profiles emerge—physical scientists display both the highest never-usage yet simultaneously topmost daily integration. Comparable daily-weekly intensiveness separates seniormost faculty (22%) underscoring an advanced career stage inflection point in committing to AI writing functionality assimilation. Taken together, AI writing tools display uneven hands-on adoption today but ChatGPT trial has partially upended this through expansive awareness building. Differences by discipline and career role provide some validating evidence for usage path dependence models hinging on exposure effects and perceived usage-enhancing utility.

**I. Measurement model evaluation.** *A. Internal consistency reliability*. Internal consistency reliability was assessed using Cronbach's alpha, rho_A, and composite reliability [72, 73]. Table 3 shows that Cronbach's alpha serves as a conservative estimate of internal consistency, whereas composite dependability represents a more generous estimate.

Intervening rho_A is optimally interpreted as the key point estimate for reliability with all three metrics having target thresholds of 0.70–0.90 for established research or exploratorily 0.60. Assessment of the outer loadings led to the elimination of several indicators that did not fulfill the minimum loading criteria. However, PB5 and AU1 were retained despite lower loadings due to theoretical considerations. The revised models therefore contain only indicators exhibiting adequate loadings (exceeding 0.70) demonstrating sufficient convergent validity per [74]. All remaining metrics are calculated based on pruned measurement models containing only satisfactory indicators. In this case, the Cronbach's alpha estimates range from 0.584 to 0.843 across the constructs, with all barring barriers exceeding the adequate 0.70 mark.

**Table 3. Assessment of internal consistency, convergent validity, and reliability for AI writing tools adoption constructs.**

| Construct | Indicators | Outer loading | Cronbach's Alpha | rho_A | Composite Reliability | Average Variance Extracted (AVE) |
|---|---|---|---|---|---|---|
| **Attitudes Towards AI Writing Tools** | AU1 | 0.699 | 0.828 | 0.842 | 0.886 | 0.662 |
| | AU2 | 0.870 | | | | |
| | AU3 | 0.851 | | | | |
| | AU4 | 0.824 | | | | |
| **Intentions to Use AI Writing Tools** | IU1 | 0.701 | 0.843 | 0.857 | 0.889 | 0.617 |
| | IU2 | 0.708 | | | | |
| | IU3 | 0.794 | | | | |
| | IU4 | 0.876 | | | | |
| | IU5 | 0.832 | | | | |
| **Perceived Barriers** | PB1 | 0.777 | 0.584 | 0.601 | 0.783 | 0.548 |
| | PB2 | 0.796 | | | | |
| | PB5 | 0.637 | | | | |
| **Perceived Subjective Norms** | PSN1 | 0.754 | 0.766 | 0.777 | 0.851 | 0.591 |
| | PSN2 | 0.850 | | | | |
| | PSN3 | 0.785 | | | | |
| | PSN5 | 0.675 | | | | |

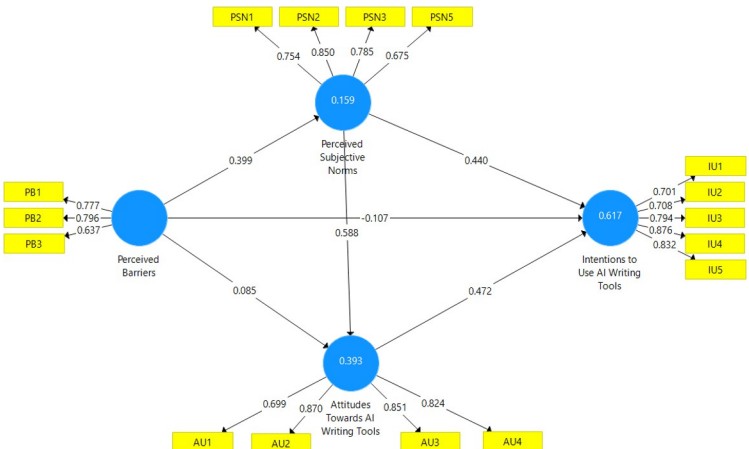

**Fig 1. Measurement model illustrating the relationships between constructs in the study.**

Composite reliability is higher as expected for all variables (from 0.783 to 0.889). Importantly, rho_A statistics surpassing respective alpha values (0.601 to 0.857) confirm better reliability assessment as point estimates per the current guidelines. Additionally, no metric exceeds 0.95 implying unlikely inflation or redundancy in internal consistency scales. In summary, cumulating evidence suggests that the measurement models demonstrate sufficient construct reliability for subsequent evaluation of the structural relationships hypothesized between them. Further validity assessments are required to fully establish adequacy (see Fig 1).

*B. Convergent validity*. Convergent validity was examined using AVE index which represents the amount of variance in indicators explained by their respective constructs [75]. Table 3 shows that values of 0.50 or higher suggest adequate convergent validity, implying that constructs account for a majority of indicator variance. In this case, AVE values range from 0.548 to 0.662 across the measurement models. Three constructs—attitudes (0.662), intentions (0.617), and norms (0.591) clearly exceed the satisfactory 0.50 threshold. This denotes that the underlying latent variables themselves explain over half the variance in their corresponding manifest indicators on average. A slightly lower AVE for barriers (0.548) indicates some potential for improvement in measurement quality to raise explanatory power. However, AVE estimates cumulatively provide evidence that the blocks demonstrate a sufficient degree of convergent validity prior to analyzing the structural model.

*C. Discriminant validity*. The HTMT ratio was used to test discriminant validity, as suggested above by traditional methodologies such as cross-loadings or the Fornell-Larcker criterion [76].

HTMT compares indicator correlations across constructs to within-construct correlations. As a proxy for discriminant validity assessment, HTMT thresholds should be under 0.85 for conceptually different constructs or 0.90 for similar ones as necessary but not sufficient conditions [77]. Table 4 shows that the HTMT statistics between the focal constructs related to AI writing tools adoption intentions range from 0.326 to 0.853, with most falling below the 0.85 standard besides norms-intentions (0.853), which still remains under the 0.90 cut-off for conceptually close constructs. This provides initial support for adequate discriminant validity and that the measures appear to be tapping into distinct latent concepts.

**II. Structural model evaluation.** *A. Collinearity assessment*. As part of the structural model evaluation, collinearity between constructs was examined using the variance inflation factor (VIF) metric [74]. The VIF quantifies the degree of correlation between predictor

**Table 4. HTMT.**

| Construct | Attitudes Towards AI Writing Tools | Intentions to Use AI Writing Tools | Perceived Barriers | Perceived Subjective Norms |
|---|---|---|---|---|
| Attitudes Towards AI Writing Tools | | | | |
| Intentions to Use AI Writing Tools | 0.834 | | | |
| Perceived Barriers | 0.449 | 0.326 | | |
| Perceived Subjective Norms | 0.781 | 0.853 | 0.609 | |

variables. Values exceeding 5 indicate critical collinearity issues, while a 3–5 range implies potential concerns per common cut-off. Table 5 shows that the outer VIFs for indicators and inner VIF statistics for predictor constructs lie between 1 and 2.5 across all exogenous variables—attitudes, barriers, and norms. No metric approached the threshold of 3. This suggests that predictors likely do not exhibit critical collinearity in the path models. Given acceptability, the subsequent hypothesis testing of structural relationships proceeds without collinearity concerns. In essence, the VIF analysis confirms the absence of discriminant validity issues between the included constructs, specifically attitudes, barriers, and norms predicting intentions. The estimated correlations between predictors are well below problematic levels. Lack of inflated correlations facilitates proceeding to evaluating the theorized causal linkages controlling for overlapping effects.

*B. Path coefficients analysis.* As outlined in Table 6 and Fig 2, bootstrapping procedures allowed testing directionality and significance of theorized predictive linkages between attitudes, barriers, norms and intentions based on recommendations by [75, 77, 78]. Two-tailed tests used 95% bias corrected accelerated confidence intervals.

H1 postulated a positive effect of favorable attitudes on AI writing tool adoption intentions. The results supported this relationship with attitudes significantly predicting intentions ($\beta = 0.472$; t = 6.323; p<0.001), thereby aligning with the initial hypothesis.

**Table 5. VIF.**

| Constructs | Indicators | VIF | Inner VIF value | | | |
|---|---|---|---|---|---|---|
| | | | Attitudes Towards AI Writing Tools | Intentions to Use AI Writing Tools | Perceived Barriers | Perceived Subjective Norms |
| Attitudes Towards AI Writing Tools | AU1 | 1.524 | | 1.647 | | |
| | AU2 | 2.343 | | | | |
| | AU3 | 1.993 | | | | |
| | AU4 | 1.873 | | | | |
| Intentions to Use AI Writing Tools | IU1 | 1.550 | | | | |
| | IU2 | 1.542 | | | | |
| | IU3 | 1.815 | | | | |
| | IU4 | 2.781 | | | | |
| | IU5 | 2.342 | | | | |
| Perceived Barriers | PB1 | 1.308 | 1.189 | 1.201 | | |
| | PB2 | 1.235 | | | | |
| | PB5 | 1.152 | | | | |
| Perceived Subjective Norms | PSN1 | 1.590 | 1.189 | 1.758 | 1.000 | |
| | PSN2 | 2.039 | | | | |
| | PSN3 | 1.661 | | | | |
| | PSN5 | 1.293 | | | | |

**Table 6. Structural model hypothesis testing outcomes.**

| | Hypothesis | β | (STDEV) | T Statistics | P Values | CIs [2.50%-97.50%] | Result |
|---|---|---|---|---|---|---|---|
| H1 | Attitudes Towards AI Writing Tools -> Intentions to Use AI Writing Tools | 0.472 | 0.075 | 6.323 | 0.000 | [0.324–0.618] | Supported |
| H2 | Perceived Barriers -> Attitudes Towards AI Writing Tools | 0.085 | 0.082 | 1.040 | 0.298 | [-0.071–0.252] | Not Supported |
| H3 | Perceived Barriers -> Intentions to Use AI Writing Tools | -0.107 | 0.072 | 1.490 | 0.136 | [-0.253–0.027] | Not Supported |
| H4 | Perceived Subjective Norms -> Attitudes Towards AI Writing Tools | 0.399 | 0.085 | 4.708 | 0.000 | [0.229–0.561] | Supported |
| H5 | Perceived Subjective Norms -> Intentions to Use AI Writing Tools | 0.588 | 0.055 | 10.696 | 0.000 | [0.477–0.694] | Supported |
| H6 | Perceived Subjective Norms -> Perceived Barriers | 0.440 | 0.076 | 5.784 | 0.000 | [0.289–0.590] | Supported |

Contrary to our expectations in H2, we found no significant link between perceived barriers and attitudes towards AI writing tools (β = 0.085; p = 0.298). This result suggests that obstacles to using AI tools may not directly shape researchers' opinions. Similarly, H3's proposal that barriers would decrease intentions to use these tools lacked statistical support (β = -0.107; p = 0.136). In other words, the challenges researchers face do not seem to directly dampen their willingness to adopt AI writing assistants.

These findings reveal a more nuanced picture of AI tool adoption than we initially assumed. The pathways we proposed from barriers to attitudes and intentions appear less straightforward than anticipated, hinting at the complexity of the factors influencing researchers' decisions to use AI writing tools.

H4 theorizing that subjective norms boost personal attitudes obtained clear evidentiary support, with social pressures positively influencing tool favorability attitudes (β = 0.399; p<0.001). Similarly, H5, stating that peer support directly heightens adoption intentions, proved significant (β = 0.588; p<0.001).

Interestingly, H6 produced an unexpected result. We originally thought that higher perceived barriers would weaken peer support for AI writing tools in academia. However, our analysis revealed a significantly positive relationship between perceived barriers and subjective norms (β = 0.440; p<0.001). This surprising outcome suggests that as researchers encounter more challenges in using AI tools, they simultaneously experience stronger social pressure to adopt them.

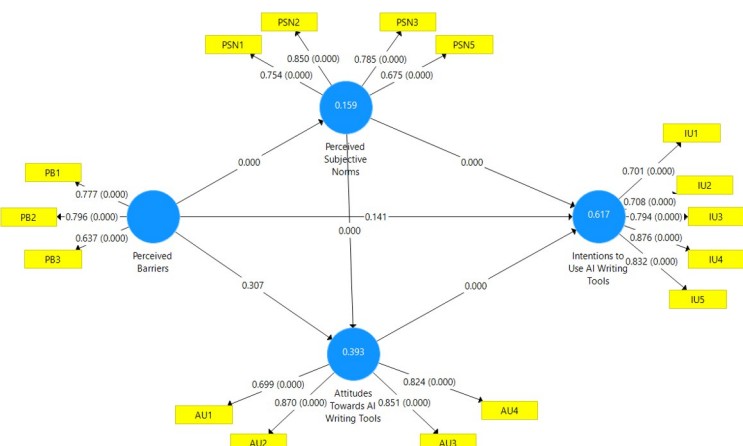

**Fig 2. Structural model showing the hypothesized pathways and their significance.**

**Table 7. Predictive capabilities analysis using PLSpredict.**

| Constructs | RMSE | MAE | $Q^2$_predict | R Square |
|---|---|---|---|---|
| Attitudes Towards AI Writing Tools | 0.973 | 0.751 | 0.077 | 0.393 |
| Intentions to Use AI Writing Tools | 1.003 | 0.801 | 0.025 | 0.617 |
| Perceived Barriers | 0.943 | 0.748 | 0.130 | 0.159 |

*C. Explanatory power assessment.* As seen in Table 7, the path model explains a substantial degree of variance in key endogenous variables 39.3% in attitudes and 61.7% in intentions towards AI-assisted writing. As [70, 79] cautions, in the absence of comparisons, raw R2 values cannot reliably indicate predictive prowess irrespective of magnitude due to potential model overfit. The scope of explained variance depends partly on model complexity in addition to fit with the phenomenon context. Yet threshold guidance for acceptable effect sizes in organizational research points to 26% as a substantial impact [80]. Hence structural relationships possess adequate explanatory power for investigator intentions and attendant attitudes. At the same time, very high values nearing 1 like in meta-analysis also require scrutiny regarding generalizability. Here too while the 61.7% hints at an excellent fit for hypothesized effects fitting the data, caution is prudent as replication with holdout samples can better establish portable predictive capacity generalizable beyond this initial theorizing.

*D. Predictive power.* As suggested by Shmueli et al [81, 82] were utilized to assess the path model's predictive prowess by comparing its out-of-sample forecast accuracy relative to a naive linear benchmark model. The PLSpredict results in Table 7 indicate lower RMSE values for key endogenous variables intentions (1.003) and attitudes (0.973), demonstrating better prediction fit versus simple linear regression models. More critically, positive Q2_predict scores for both constructs confirm predictive relevance, satisfying the threshold for model prognostic superiority over using mere mean values. Together, these metrics offer initial evidence of acceptable out-of-sample predictive quality beyond just in-sample fit. The structural relationships demonstrate transportable predictive capacity generalizable to the research context.

*E. Model fit.* Rather than GoF, model fit was examined using SRMR along with a bootstrap-based 95% CI test as suggested by Schuberth [83]. The sample size (~150) allows such an analysis, unlike complex models. Table 8 shows that the model fit analysis centers on the Standardized Root Mean Square Residual (SRMR) metric. The original SRMR value of 0.093 exceeds the threshold of 0.08 for establishing model fit. This initial result implies a potential lack of fit between the conceptual model and observed data. However, further analysis using bootstrapping provides an additional perspective. The bootstrap-derived mean SRMR was notably lower at 0.064. Additionally, the 95% confidence interval lower limit from resampling is 0.074, which is very close to the 0.08 cutoff. This finding suggests that the model plausibility cannot be strictly ruled out based solely on the original estimate.

While the SRMR values slightly exceed the conventional 0.08 threshold [84], we argue that this moderate fit is acceptable in the context of our study. As pioneering research applying the Theory of Reasoned Action to AI writing tool adoption in academia, our model represents an initial exploration of a complex phenomenon. In such exploratory contexts [85], suggest that slightly higher SRMR values can be tolerated.

**Table 8. Validation of model fit.**

| | SRMR | Sample Mean (M) | 95% | 99% |
|---|---|---|---|---|
| Saturated Model | 0.093 | 0.064 | 0.074 | 0.079 |
| Estimated Model | 0.093 | 0.064 | 0.074 | 0.079 |

The consistency of SRMR values between the saturated and estimated models (both 0.093) indicates that our theoretical model captures meaningful data relationships [60]. Our bootstrap analysis further supports the model's robustness, with a mean SRMR of 0.064 and a 95% confidence interval lower limit of 0.074 both near or below the 0.08 threshold. This aligns with [86] assertion that bootstrap results can provide additional support for the model fit.

Moreover, our model explains substantial variance in key outcomes (61.7% for adoption intentions), demonstrating its practical utility. As [87] argue, explanatory power is a crucial consideration alongside fit indices in evaluating model adequacy. Given the complexity of AI tool adoption in academia and our sample size considerations (n = 150), achieving a perfect fit is challenging. [88] cautions against overly rigid adherence to fit thresholds, especially in complex fields and with moderate sample sizes.

Rather than dismissing outright misfit unilaterally early on, we adopt a measured position maintaining inconclusive evidence towards model's plausible potential to reproduce meaningful patterns in phenomenon relationships that withstands bootstrap scrutiny. As this study represents initial empirical testing of AI writing tools adoption through a reasoned action lens, accepting potential adequate approximations seems reasonable.

## Discussion

The findings of this study provide valuable insights into the factors influencing academic researchers' adoption of AI writing tools, grounded in TRA framework. The results offer both support for and contradictions with previous evidence in the literature, highlighting the unique context of AI writing tool adoption in academia.

One of the key findings is the strong positive influence of favorable attitudes on researchers' intentions to use AI writing tools (H1), which aligns with the core propositions of the TRA [24] and is consistent with prior research on technology adoption in various contexts [27, 35]. This suggests that researchers who hold positive evaluations of AI writing tools, based on their perceived benefits and outcomes, are more likely to intend to adopt these technologies in their academic writing processes.

However, the non-significant effect of perceived barriers on attitudes (H2) and intentions (H3) contradicts some previous findings in the literature. Studies have shown that perceived barriers, such as technical difficulties, ethical concerns, and institutional restrictions, can negatively impact attitudes and adoption intentions towards AI technologies [51, 52]. The lack of significant effects in this study may be due to the specific nature of AI writing tools and the academic context, where researchers may prioritize potential benefits over barriers or have strategies to overcome these barriers. This intriguing finding diverges from previous research on technology adoption [51, 52] and may be attributed to several factors specific to the nature of AI writing tools and the strategies researchers employ to overcome barriers. Firstly, AI writing tools are designed to enhance productivity and efficiency in the writing process [1], which may lead researchers to prioritize potential benefits over perceived barriers. Secondly, researchers may have developed strategies to mitigate the impact of barriers, such as seeking institutional support, collaborating with peers, or investing in their own skill development [68]. These strategies could reduce the influence of barriers on attitudes and intentions. Thirdly, rapid advancements in AI writing technologies may have created a perception among researchers that barriers are temporary and will be resolved as the tools mature [3]. To further investigate these potential explanations, future research could examine the specific strategies researchers use to overcome barriers and how the evolving nature of AI writing tools influences perceptions of barriers over time. Additionally, qualitative studies could provide valuable

insights into the thought processes and experiences of researchers as they navigate the adoption of AI writing tools in the face of perceived barriers.

The significant positive influence of subjective norms on attitudes (H4) and intentions (H5) highlights the crucial role of social influence in shaping researchers' perceptions and adoption decisions regarding AI writing tools. This finding is consistent with prior research that has demonstrated the importance of peer influence and social pressure in technology adoption [35, 53]. In the academic context, the opinions and expectations of colleagues, supervisors, and the broader research community can significantly shape researchers' attitudes and intentions towards AI writing tools. Interestingly, the study found a significant negative connection between subjective norms and perceived barriers (H6), suggesting that when influential others in the academic community express support for and encourage the use of AI writing tools, researchers may perceive fewer barriers to adoption. This finding extends previous research by highlighting the potential of social influence to mitigate perceived barriers in the adoption process [53].

The explanatory power of the TRA model in this study, with 39.3% of the variance in attitudes and 61.7% of the variance in intentions explained, is comparable to or higher than that reported in previous studies applying the TRA to technology adoption contexts [30, 31]. This suggests that the TRA provides a robust framework for understanding AI writing tool adoption among academic researchers. However, the relatively lower explanatory power for attitudes compared to intentions indicates that there may be additional factors beyond subjective norms and perceived barriers that shape researchers' evaluations of AI writing tools, such as individual differences or tool-specific characteristics. The predictive power of the model, as assessed using PLSpredict. [81, 82, 89] demonstrates the model's ability to generate accurate out-of-sample predictions for attitudes and intentions. This finding strengthens the validity and generalizability of the TRA framework in the context of AI writing tool adoption, suggesting that the relationships identified in this study may hold true for researchers beyond the current sample. While the model fit indices (SRMR) initially suggested a potential lack of fit, the bootstrap-based analysis provided a more nuanced perspective, indicating that the model's plausibility cannot be entirely ruled out. As this study represents an initial empirical test of AI writing tool adoption through the TRA lens, the results provide a foundation for further research and model refinement.

In contextualizing our TRA model's performance within the broader landscape of AI adoption research, we find both encouraging and intriguing similarities. Our model's explanatory power stands out, accounting for 39.3% of the variance in attitudes and an impressive 61.7% in adoption intentions towards AI writing tools. This performance aligns well with recent studies, suggesting the robustness of our approach.

For instance [13], in their study of AI-enabled robot advisors, reported explanatory power ranging from 41.2% to 67.7% for various adoption-related constructs. Our model's ability to explain 61.7% of the variance in intentions falls comfortably within this range, despite our more focused use of the TRA compared to their combined UTAUT and TRA approach. This suggests that our streamlined model performs comparably to more complex frameworks in capturing the essence of AI adoption intentions.

The strength of our model is further evident in the significant relationships that we uncovered. We found that both attitudes ($\beta = 0.472$, $p < 0.001$) and subjective norms ($\beta = 0.588$, $p < 0.001$) strongly influence adoption intentions. These findings resonate with those of Cho et al. (2023), who also applied TRA to examine ChatGPT adoption. While they didn't report specific coefficients, their results similarly highlighted the crucial role of attitudes and subjective norms in shaping adoption intentions.

Interestingly, our study diverges from some previous research in its findings regarding perceived barriers. We found no significant effect of barriers on either attitudes ($\beta = 0.085$, $p = 0.298$) or intentions ($\beta = -0.107$, $p = 0.136$). This contrasts with studies like [13], where factors akin to barriers, such as perceived security and privacy, significantly influenced attitudes and adoption intentions. This difference underscores the unique context of AI writing tools in academia, suggesting that the role of barriers may differ substantially from other domains, such as financial technologies.

In terms of model fit, our SRMR of 0.093 sits slightly above the ideal threshold of 0.08. However, this is not uncommon in the field. Many studies using structural equation modeling or PLS-SEM in AI adoption research report SRMR values between 0.05 and 0.10, placing our model within an acceptable range. This suggests that while there's room for improvement, our model fit is comparable to others in the field.

A unique strength of our study was its assessment of predictive power. Using PLSpredict, we found positive $Q^2$_predict scores for both attitudes (0.077) and intentions (0.025). Few studies in the AI adoption domain report this metric, making our contribution particularly valuable. These positive values indicate our model's predictive relevance, showcasing its ability to generate accurate out-of-sample predictions, a crucial aspect often overlooked in similar research.

The context of our study–AI writing tools in academia–also sets this apart. While researchers such as [11, 89] have explored AI adoption in academic settings, they have employed different theoretical frameworks such as Diffusion of Innovation and TAM. Our application of the TRA in this specific context offers novel insights, particularly in highlighting the strong role of subjective norms in academic AI tool adoption.

Hence, the performance of our TRA model stands up well when compared to recent studies on AI adoption. Its high explanatory power for adoption intentions, the significant influence of attitudes and subjective norms, and its demonstrated predictive relevance underscore the robustness of the TRA in understanding AI writing tool adoption in academia. Simultaneously, our unexpected findings regarding perceived barriers raise new questions about the nature of adoption challenges in academic contexts. This blend of confirmatory results and novel insights positions our study as a valuable contribution to the evolving landscape of research on AI adoption, particularly in the realm of academic tools and practices.

While this study provides valuable insights into the factors influencing AI writing tool adoption among academic researchers, it is important to acknowledge its limitations. The cross-sectional design limits the ability to establish causal relationships between variables, and future research could employ longitudinal designs to examine the temporal dynamics of the adoption process. Additionally, while the sample is diverse, larger and more heterogeneous samples could further enhance the generalizability of the findings. Despite these limitations, the study's strengths lie in its application of the well-established TRA framework, its diverse sample, and its use of rigorous statistical analyses. Moreover, the findings lay the groundwork for future research to explore additional factors, such as individual differences, tool-specific characteristics, and contextual factors, to develop a more comprehensive understanding of AI writing tool adoption in academia.

## Implications

Our study on the adoption of AI writing tools among academic researchers yields significant implications for both theory and practice. These insights offer valuable guidance for various stakeholders in the academic community and beyond.

## Theoretical implications

From a theoretical standpoint, our study extends the application of the Theory of Reasoned Action (TRA) into the novel domain of AI writing tools in academia. This extension demonstrates the theory's robustness in explaining adoption intentions in emerging technological contexts. Particularly intriguing is our finding regarding the non-significant effect of perceived barriers, which challenges conventional wisdom in the technology adoption literature. This result suggests a need for a more nuanced understanding of barriers in specialized, innovation-driven environments, such as academia. This opens up new avenues for research into how different contexts might alter the impact of perceived barriers on technology adoption.

Furthermore, our study underscores the critical role of social influence in the adoption of academic technology. The strong influence of subjective norms on both attitudes and intentions highlights that in academic settings, the opinions and behaviors of peers and mentors may carry even more weight than in other professional contexts. This finding enriches our understanding of how social dynamics operate in knowledge-intensive environments and their impact on technology adoption decisions.

## Practical implications

Turning to practical implications, our findings offer actionable insights for various stakeholders involved in the development and implementation of AI writing tools in academia.

For tool developers, the key lies in emphasizing the benefits of their products while enhancing the user experience. By clearly communicating the productivity gains and quality improvements offered by AI writing tools, developers can foster positive attitudes among potential users. For instance, implementing features that quantify the time saved or demonstrate writing quality enhancements could provide tangible evidence of the tool's benefits. Additionally, creating user-friendly interfaces and comprehensive tutorials can address potential technical barriers, making the tools more accessible to a wider range of users. Developers should also consider incorporating features that facilitate peer learning and collaboration, as our study highlights the importance of social influence in adoption decisions.

Academic institutions play a crucial role in shaping the adoption landscape. They should focus on developing clear ethical guidelines for the use of AI writing tools in academic work, addressing concerns about integrity and appropriate use. Organizing workshops and training sessions can familiarize faculty and students with these tools and address skill-related barriers. Moreover, fostering a supportive environment that encourages the exploration and responsible adoption of AI writing tools can leverage the power of social influence highlighted in our study. Initiatives such as an "AI Writing Tools Champions" program could create a network of early adopters who can mentor colleagues and showcase successful applications.

Publishers and journal editors are at the forefront of academic communication and can significantly influence adoption trends. Developing clear policies regarding the use of AI writing tools in manuscript preparation and submission is crucial. This could include introducing standardized declaration forms for authors to transparently report their use of AI tools. Publishers might also consider integrating AI writing tool features into their submission platforms, normalizing their use in the publication process. Additionally, providing resources and guidance to reviewers on how to evaluate manuscripts that may have been created with AI assistance can ensure fair and informed assessment of such work.

For individual researchers, our findings suggest a gradual approach to adoption. Starting with AI writing tools in low-stakes writing tasks can help build familiarity and positive attitudes. Engaging in peer discussions about these tools can contribute to positive subjective norms within academic circles. Researchers should also develop personal strategies to address

perceived barriers, such as creating checklists for ethical AI use in writing that include steps for fact-checking and ensuring original contributions.

By implementing these strategies, stakeholders can foster an environment that promotes the responsible and effective adoption of AI writing tools in academia. This approach leverages the key factors identified as drivers of adoption intentions: positive attitudes, strong subjective norms, and strategies to overcome perceived barriers. As AI continues to evolve and shape the future of scholarly communication, understanding and acting upon these implications will be crucial for harnessing the potential of AI writing tools while ensuring their responsible and effective use in academic contexts.

This study provides valuable insights into the factors influencing academic researchers' adoption of AI writing tools, highlighting the roles of attitudes, subjective norms, and perceived barriers within the TRA framework. The findings contribute to the literature on technology adoption in professional contexts and offer practical implications for stakeholders involved in the development and implementation of AI writing tools in academia. As AI technologies continue to evolve and shape the future of scholarly communication, understanding the drivers and barriers to their adoption will be crucial for harnessing their potential while ensuring responsible and effective use.

## Conclusion

This study represents a significant step forward in understanding the factors that influence academic researchers' adoption of AI writing tools. By applying the Theory of TRA framework, the study has provided valuable insights into the roles of attitudes, subjective norms, and perceived barriers in shaping researchers' intentions to adopt these technologies.

Our findings reveal a nuanced picture of AI writing tool adoption in academia. Favorable attitudes towards these tools emerged as a strong predictor of adoption intentions, underscoring the importance of promoting positive perceptions based on the tools' perceived benefits and outcomes. The role of social influence in the adoption process is equally crucial. We found that subjective norms, encompassing the opinions and expectations of colleagues, supervisors, and the broader research community, significantly shape both researchers' attitudes and their intentions to adopt AI writing tools.

Interestingly, and contrary to some previous technology adoption research, perceived barriers did not demonstrate a significant direct effect on attitudes or intentions in our study. This unexpected finding suggests that in the academic context, the potential benefits of AI writing tools may outweigh perceived obstacles. Moreover, it hints at the possibility that social influence might play a role in mitigating perceived barriers to adoption, offering a valuable insight for stakeholders aiming to promote the responsible and effective use of these tools in academia.

These results suggest several actionable strategies for key stakeholders. Tool developers should focus on clearly demonstrating the benefits and productivity gains of their products, potentially through features that quantify the time saved or writing quality improvements. Academic institutions could leverage the power of social influence by developing comprehensive training programs and establishing "AI Writing Tool Champions." Publishers, as gatekeepers of academic communication, might consider creating clear guidelines for AI tool use in manuscript preparation and integrating AI-assisted writing features into submission platforms to normalize their use.

Our study makes a noteworthy contribution to the literature on technology adoption in professional contexts. By highlighting the importance of domain-specific factors and the unique characteristics of AI writing tools, we have enriched the understanding of adoption

intentions and behaviors in specialized settings. The strong explanatory power of the TRA model in this study, coupled with its demonstrated predictive power using PLSpredict, affirms the robustness of this framework for understanding AI writing tool adoption among academic researchers.

While acknowledging the study's limitations, including its cross-sectional design and reliance on self-reported measures, we see them as springboards for future research. Longitudinal studies could track changes in attitudes and adoption behaviors over time. Investigations into additional variables such as writing task complexity, discipline-specific norms, and individual traits could provide a more nuanced understanding. Mixed-methods approaches and experimental studies could offer deeper insights into researchers' decision-making processes and the effectiveness of different adoption strategies. Cross-cultural comparisons could illuminate how cultural factors influence AI writing tool adoption across diverse academic contexts.

As AI technologies continue to evolve and transform the landscape of academic writing, understanding the drivers and barriers to their adoption remains crucial. This study lays the groundwork for further research and provides actionable insights for stakeholders involved in developing and implementing AI writing tools in academia. Ongoing collaboration and dialogue among researchers, tool developers, and academic institutions will be essential to navigate the ethical, social, and practical implications of these tools.

By fostering a culture of responsible innovation and adoption, the academic community can harness the power of AI to enhance scholarly productivity and communication while addressing potential challenges and concerns. As this field progresses, future research should build on these findings, expand our understanding, and develop more comprehensive models to facilitate the appropriate and efficient integration of AI technologies into academic writing practices. In doing so, we can work towards a future where AI writing tools serve as valuable allies in the pursuit of knowledge and scholarly excellence.

## Supporting information

**S1 Data. Dataset of the study, containing all data collected and used for analysis, including variables on AI tool adoption and researcher demographics.**
(XLSX)

**S1 Appendix. Questionnaire used for data collection, detailing all survey items related to the adoption of AI writing tools among academic researchers.**
(DOCX)

## Author Contributions

**Conceptualization:** Mohammed A. Al-Bukhrani, Yasser Mohammed Hamid Alrefaee, Mohammed Tawfik.

**Data curation:** Mohammed Tawfik.

**Formal analysis:** Mohammed A. Al-Bukhrani.

**Methodology:** Mohammed A. Al-Bukhrani, Yasser Mohammed Hamid Alrefaee.

**Supervision:** Yasser Mohammed Hamid Alrefaee.

**Writing – original draft:** Mohammed A. Al-Bukhrani.

**Writing – review & editing:** Yasser Mohammed Hamid Alrefaee, Mohammed Tawfik.

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
