## [Decision Letter · Decision Letter 0]

10 Sep 2024

PONE-D-24-17396Adoption of AI Writing Tools among Academic Researchers: A Theory of Reasoned Action ApproachPLOS ONE

Dear Dr. A-lbukhrani,

Thank you for submitting your manuscript to PLOS ONE. After careful consideration, we feel that it has merit but does not fully meet PLOS ONE’s publication criteria as it currently stands. Therefore, we invite you to submit a revised version of the manuscript that addresses the points raised during the review process.

**ACADEMIC EDITOR**

After a thorough review, some concerns regarding the manuscript need to be addressed. Based on my assessment, the manuscript contains frequent grammatical errors, awkward phrasing, and some clarity issues that require attention. While the issues observed may not necessitate a complete overhaul, they do suggest that significant revisions are needed to enhance the overall quality of the manuscript. The primary concerns are related to fine-tuning rather than structural changes. For your reference, I have attached a document highlighting specific areas where improvements can be made. This includes detailed suggestions for improving grammar, phrasing, and clarity. I believe addressing these issues will greatly contribute to the manuscript’s readability and effectiveness.

Thank you for your attention to these matters. I look forward to seeing the revised version of the manuscript.

Best regards,

We look forward to receiving your revised manuscript.

Kind regards,

Dokun Iwalewa OIuwajana

Academic Editor

PLOS ONE

Journal Requirements:

1. When submitting your revision, we need you to address these additional requirements. Please ensure that your manuscript meets PLOS ONE's style requirements, including those for file naming. The PLOS ONE style templates can be found at https://journals.plos.org/plosone/s/file?id=wjVg/PLOSOne_formatting_sample_main_body.pdf and https://journals.plos.org/plosone/s/file?id=ba62/PLOSOne_formatting_sample_title_authors_affiliations.pdf 2. Please provide additional details regarding participant consent. In the ethics statement in the Methods and online submission information, please ensure that you have specified (1) whether consent was informed and (2) what type you obtained (for instance, written or verbal, and if verbal, how it was documented and witnessed). If your study included minors, state whether you obtained consent from parents or guardians. If the need for consent was waived by the ethics committee, please include this information. If you are reporting a retrospective study of medical records or archived samples, please ensure that you have discussed whether all data were fully anonymized before you accessed them and/or whether the IRB or ethics committee waived the requirement for informed consent. If patients provided informed written consent to have data from their medical records used in research, please include this information. 3. Please provide a complete Data Availability Statement in the submission form, ensuring you include all necessary access information or a reason for why you are unable to make your data freely accessible. If your research concerns only data provided within your submission, please write "All data are in the manuscript and/or supporting information files" as your Data Availability Statement.

Additional Editor Comments:

Dear authors, based on the frequent grammatical errors, awkward phrasing, and significant clarity issues in the manuscript, major revisions might be necessary. However, the overall assessment suggests that the issues are more related to fine-tuning rather than requiring a complete overhaul. Please see the attached document for specific suggestions on how to improve the manuscript. Thank you.

Reviewers' comments:

Reviewer's Responses to Questions

**Comments to the Author**

1. Is the manuscript technically sound, and do the data support the conclusions?

Reviewer #1: Yes

2. Has the statistical analysis been performed appropriately and rigorously? 

Reviewer #1: Yes

3. Have the authors made all data underlying the findings in their manuscript fully available?

Reviewer #1: Yes

4. Is the manuscript presented in an intelligible fashion and written in standard English?

Reviewer #1: Yes

5. Review Comments to the Author

Reviewer #1: I have conducted a comprehensive review of your paper and found it to have some merit. However, several key areas require further improvement before publication. I have summarized the necessary changes, hoping that the feedback will be valuable to you as you revise the paper. I cannot consider your manuscript for publication now, but I encourage you to consider the feedback and resubmit your revised manuscript.

Introduction

- The introduction relies heavily on the perceived hype around AI tools without providing sufficient empirical evidence to support the claims. Including more concrete examples and data on the current adoption rates of AI tools would be beneficial. Additionally, while the introduction mentions a research gap, it does not delineate which aspects of AI writing tool adoption are under-researched. Clarifying this would help establish the study's significance more robustly. Importantly, a more focused purpose statement would provide clearer direction for the research and its contributions.

Literature Review and Hypotheses Development

The review of emerging AI writing technologies is quite general. It would benefit from a deeper exploration of specific tools, their functionalities, and their impact on scholarly work.

The literature review presents various studies and models but lacks a critical analysis of their strengths and limitations. A more nuanced discussion would provide a better foundation for the study. Additionally, while the Theory of Reasoned Action (TRA) is explained, the justification for choosing this model over others like TAM or UTAUT is not compelling. Providing a stronger rationale based on the unique aspects of AI tool adoption in academia would strengthen this section. The explanation of TRA is quite general, and more specific examples of how TRA has been successfully applied in similar contexts would be helpful. The discussion of other models (TAM, UTAUT) is brief and does not delve into their specific components and how they compare to TRA. There is no critical comparison of the different models to highlight why TRA was chosen. Discussing the relative strengths and weaknesses of each model would add value. Some hypotheses are broad and could benefit from more specificity. For instance, instead of "Perceived barriers negatively influence researchers' attitudes," specifying which barriers (technical, ethical, etc.) would provide more clarity. The hypotheses are stated without sufficiently linking them back to the theoretical constructs of TRA, and stronger theoretical underpinning would make the hypotheses more compelling.

Methodology

The explanation of purposive sampling is somewhat vague. Providing more details on the criteria for selecting participants would enhance understanding. Although the sample size determination is mentioned, the rationale for the chosen effect size and power level is not thoroughly explained. Providing more context would clarify why these thresholds were selected. The methods of participant recruitment are briefly mentioned but lack detail. For instance, how were the email addresses obtained, and what specific strategies were used to ensure a diverse and representative sample? The process of adapting existing scales is mentioned, but more detail on the specific modifications made would be helpful. This includes providing examples of adapted items. Also, since this is a self-reported questionnaire, how did you establish if there is social desirability bias in the data collected?

- Articulate recommendations for practitioners to enhance the application of the research.

A section should be included to discuss limitations and future work.

6. PLOS authors have the option to publish the peer review history of their article (what does this mean?). If published, this will include your full peer review and any attached files.

Reviewer #1: **Yes: **Musa Adekunle Ayanwale

---

## [Author Response · Author response to Decision Letter 0]

25 Sep 2024

Dear

We sincerely appreciate the time and effort you and the reviewers have dedicated to providing valuable feedback on our manuscript titled "Adoption of AI Writing Tools among Academic Researchers: A Theory of Reasoned Action Approach" (PONE-D-24-17396). We have carefully considered all the comments and suggestions, and have made substantial revisions to address them. We believe these changes have significantly improved the quality and clarity of our manuscript.

In response to the feedback received, we have made the following key improvements:

1. Overall manuscript quality: We have thoroughly revised the entire manuscript to address grammatical errors, awkward phrasing, and clarity issues. This includes a comprehensive grammar and spell check, revision of phrasing, and review.

2. Introduction and Literature Review: We have clarified the unique contributions of TRA compared to UTAUT, added recent statistics on AI writing tools usage in academia, and provided clearer research questions. We've also expanded our justification for using TRA and included more examples of its application in AI adoption contexts.

3. Methodology: We have provided more detailed rationales for our methodological choices, including sampling strategy and sample size. We've also added information on scale adaptation and data quality procedures.

4. Results and Discussion: We have improved the clarity of our results presentation, including a clearer interpretation of non-significant results. We've also expanded our discussion to include more detailed comparisons with recent AI adoption studies and more specific implications for stakeholders.

5. Conclusion: We have restructured the conclusion to more clearly state our key findings and provide more specific recommendations and future research directions.

For a detailed point-by-point response to each comment, please refer to the attached table. We believe that these revisions have substantially improved our manuscript and addressed all the concerns raised. We have also highlighted all changes made to the original version.

Thank you again for your valuable feedback and the opportunity to revise our manuscript. We look forward to your response and are happy to address any further questions or concerns you may have.

Sincerely, 

Mohammed A. Al-bukhrani

Corresponding author

---

## [Decision Letter · Decision Letter 1]

1 Nov 2024

Adoption of AI Writing Tools among Academic Researchers: A Theory of Reasoned Action Approach

PONE-D-24-17396R1

Dear Dr.
Mohammmed A-lbukhrani,

We’re pleased to inform you that your manuscript has been judged scientifically suitable for publication and will be formally accepted for publication once it meets all outstanding technical requirements.

Kind regards,

Dokun Iwalewa OIuwajana

Academic Editor

PLOS ONE

Additional Editor Comments (optional):

Reviewers' comments:

Reviewer's Responses to Questions

**Comments to the Author**

1. If the authors have adequately addressed your comments raised in a previous round of review and you feel that this manuscript is now acceptable for publication, you may indicate that here to bypass the “Comments to the Author” section, enter your conflict of interest statement in the “Confidential to Editor” section, and submit your "Accept" recommendation.

Reviewer #1: All comments have been addressed

Reviewer #2: (No Response)

2. Is the manuscript technically sound, and do the data support the conclusions?

Reviewer #1: Yes

Reviewer #2: (No Response)

3. Has the statistical analysis been performed appropriately and rigorously? 

Reviewer #1: Yes

Reviewer #2: (No Response)

4. Have the authors made all data underlying the findings in their manuscript fully available?

Reviewer #1: Yes

Reviewer #2: (No Response)

5. Is the manuscript presented in an intelligible fashion and written in standard English?

Reviewer #1: Yes

Reviewer #2: (No Response)

6. Review Comments to the Author

Reviewer #1: The revisions you have made have significantly improved the clarity and depth of the manuscript. I appreciate the thoughtful responses provided and the comprehensive improvements across the grey areas identified earlier. Thank you.

Reviewer #2: (No Response)

7. PLOS authors have the option to publish the peer review history of their article (what does this mean?). If published, this will include your full peer review and any attached files.

Reviewer #1: **Yes: **Musa Adekunle Ayanwale

Reviewer #2: No

---

## [Editor Report · Acceptance letter]

8 Nov 2024

PONE-D-24-17396R1 

PLOS ONE

Dear Dr. Al-Bukhrani, 

I'm pleased to inform you that your manuscript has been deemed suitable for publication in PLOS ONE. Congratulations! Your manuscript is now being handed over to our production team.

Kind regards, 

on behalf of

Asst Dokun Iwalewa OIuwajana 

Academic Editor

PLOS ONE